# De novo generation of multi-target compounds using deep generative chemistry

Brenton P. Munson[1,2], Michael Chen[1], Audrey Bogosian[1], Jason F. Kreisberg [1], Katherine Licon[1], Ruben Abagyan[3], Brent M. Kuenzi[1] & Trey Ideker [1,2,4] ✉

Polypharmacology drugs—compounds that inhibit multiple proteins—have many applications but are difficult to design. To address this challenge we have developed POLYGON, an approach to polypharmacology based on generative reinforcement learning. POLYGON embeds chemical space and iteratively samples it to generate new molecular structures; these are rewarded by the predicted ability to inhibit each of two protein targets and by drug-likeness and ease-of-synthesis. In binding data for >100,000 compounds, POLYGON correctly recognizes polypharmacology interactions with 82.5% accuracy. We subsequently generate *de-novo* compounds targeting ten pairs of proteins with documented co-dependency. Docking analysis indicates that top structures bind their two targets with low free energies and similar 3D orientations to canonical single-protein inhibitors. We synthesize 32 compounds targeting MEK1 and mTOR, with most yielding >50% reduction in each protein activity and in cell viability when dosed at 1–10 μM. These results support the potential of generative modeling for polypharmacology.

Classical drug discovery operates by a "one disease—one target—one drug" model. While this model has yielded numerous successful therapies, many diseases do not have a single molecular cause but instead are associated with various potential points of intervention, each of which may have a partial effect on disease etiology. Such difficulty is especially apparent for polygenic diseases like cancer and psychiatric disorders, which integrate functional effects across many genes organized in complex biological networks[1–3].

Accordingly, there has been increasing interest in treatment strategies that address multiple targets[4]. This strategy can be achieved through a combination of multiple therapies or through the use of polypharmacology drugs, which bind and functionally modulate two or more molecular targets simultaneously. While polypharmacology is still in its infancy, recent studies have begun to demonstrate its utility in treating disease. For example, several groups have shown that KRAS mutant non-small cell lung cancers, which have been recalcitrant to treatment with classical single-target agents, can be effectively treated using polypharmacological compounds[5–7]. Polypharmacology also offers potential advantages over combination therapy, such as a superior pharmacokinetic and safety profile, lower likelihood of acquired resistance, and simplified therapy formulation leading to increased patient compliance[4,8,9].

A major barrier to polypharmacology compounds has been the challenge of designing a single agent that potently inhibits multiple proteins simultaneously[8]. Effective polypharmacology designs, such as agents targeting RET and VEGFR2 in thyroid cancer[10], required substantial time and resources before suitable hit scaffolds could be identified. For these reasons, such compounds have largely been discovered serendipitously rather than systematically[5,11]. On the other hand, many recent advancements in machine learning are beginning to show promise in related activities, including systematic prediction of compound-target interactions[12], de novo generation of single-target inhibitors[13–18], and recognition of existing drugs with polypharmacology (dual targeting) potential[19].

[1]Division of Human Genomics and Precision Medicine, Department of Medicine, University of California San Diego, La Jolla, CA 92093, USA. [2]Department of Bioengineering, University of California San Diego, La Jolla, CA 92093, USA. [3]Skaggs School of Pharmacy and Pharmaceutical Sciences, University of California San Diego, La Jolla, CA 92093, USA. [4]Department of Computer Science and Engineering, University of California San Diego, La Jolla, CA 92093, USA. ✉e-mail: tideker@health.ucsd.edu

To enable the programmatic generation of new poly-pharmacology compounds, we developed the POLYpharmacology Generative Optimization Network (POLYGON), a deep machine learning model based on generative AI and reinforcement learning[13–18]. In what follows, we first describe the architecture of POLYGON and how it was trained and benchmarked. Using this model, we then generate a collection of de novo molecular compounds targeting ten pairs of synthetically lethal cancer proteins, which we assess by several means including 3D molecular docking analysis. Finally, we synthesize 32 POLYGON compounds generated for dual inhibition of MEK1 and mTOR, which we subsequently validate in both cell-free assays and lung tumor cells.

## Results and discussion

### A generative model for polypharmacology

Akin to de novo generation of human portraits[20], where multiple independent facial features can be tuned toward a specific goal (e.g. mood and age), POLYGON optimizes multiple, potentially independent, chemical properties (Fig. 1). The first component of POLYGON is a variational autoencoder[21] (VAE), a type of deep neural network, which is used to process the chemical formula of a molecular compound into a "chemical embedding" (Fig. 2a). An embedding is a low-dimensional representation of a complex input, in which each data point (here a chemical structure) is assigned coordinates in the reduced dimensions. Much machine learning research has focused on creating a good embedding, where similar inputs (here, similar chemical structures) are close in the embedded space[22]. This chemical encoder is coupled to a decoder, which converts any position in the chemical embedding back into a valid molecular formula (Fig. 2a).

This encoder/decoder model was trained (Supplementary Fig. 1a) using a diverse dataset of over one million small molecules drawn from the ChEMBL database[15,23] (Methods). Once trained, we verified the model was able to encode and recover the chemical formulas of held-out molecules that had not been used for training (Supplementary Fig. 1b). A further important aspect for de novo molecule generation is the ability to decode any coordinate in the embedding into a valid chemical compound. In this respect, we found that most coordinates selected from the chemical embedding resulted in valid new SMILES strings (Supplementary Fig. 1c). We also examined the extent to which compounds drawn from similar positions in the chemical embedding were able to bind the same targets. For this purpose, we amassed

18,763 compound-target binding affinities measured for 24 different targets[24], drawing from sources including BindingDB[25] and Pharos[26,27]. We observed that pairs of compounds with affinity for the same target were significantly closer in the chemical embedding than compounds with affinity for different targets ($p < 0.01$; one-sided $t$ test = −50.2; DOF = 5,757,396; 95% CI −9.12 to −0.10; $n = 100$ compounds per target tested). Multiclass target prediction for held-out compounds had individual target accuracies ranging from 0.76 to 0.95 (Area under Receiver Operating Characteristic, Supplementary Fig. 1d) with an accuracy of $0.85 \pm 0.05$ (mean ± stdev). Some targets had partial overlap in their corresponding distributions of compounds (Fig. 2b, Supplementary Fig. 2), suggesting regions of the embedding with potential for polypharmacology.

This embedding framework was used to control the second major component of POLYGON, a reinforcement learning system for the generation of polypharmacology compounds with activity against two different targets of interest (Fig. 2c). Reinforcement learning[15,28] is a powerful machine learning strategy by which a model is trained iteratively, at each step rewarding desired outputs and/or punishing undesired ones. It differs from supervised learning in that it does not need labeled input/output pairs but instead seeks a balance between exploitation of current knowledge and exploration of uncharted territory (here, chemical space)[29]. In our implementation, compounds were randomly sampled from the chemical embedding and scored based on their predicted ability to inhibit each of two specific targets, together with multiple other reward criteria related to compound synthesizability and drug-likeness (Methods)[30]. Coordinates of high-scoring compounds were then used to define reduced subspaces of the chemical embedding for model retraining and random sampling in further iterations (Fig. 2c, Methods), yielding compounds of increasingly high quality. Predictions of the compound-target scoring module compared favorably to those of previous compound-target predictors, ranking in the top tier of those evaluated in a recent IDG-DREAM competition (Supplementary Fig. 3, Methods). As another benchmark, we tasked POLYGON with scoring a held-out set of (compound, target 1, target 2) triplets for which the IC50s against both targets had been characterized and recorded in BindingDB (covering a broad set of 109,811 compounds and 1850 targets; Fig. 3a). At an activity threshold of IC50 < 1 μM, POLYGON achieved an accuracy of 81.9% in classifying cases for which the compound was active against both targets, i.e. showed polypharmacologic activity ($p = 2.2 \times 10^{-16}$; 95% CI 20.7 to 22.0;

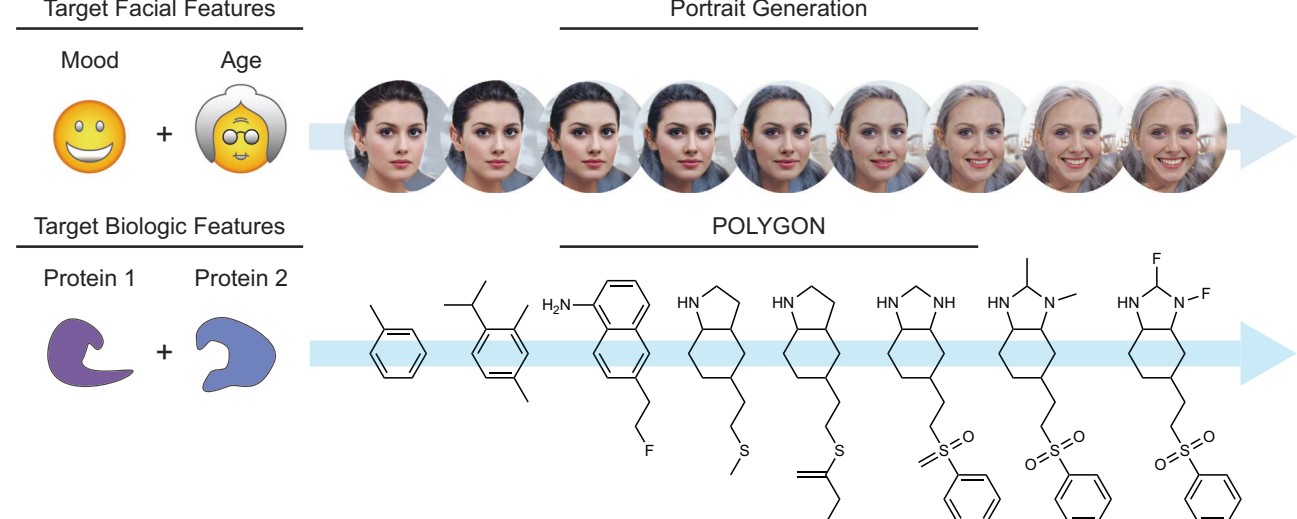

**Fig. 1 | Generative models of portraits versus small molecules.** Multiple facial features such as mood and age can be simultaneously optimized in generating de novo portraits of faces (top). Likewise, multiple biological features such as inhibition of two protein targets can be simultaneously optimized in generating de novo small molecules (bottom). Portrait images were generated with StarGAN v2[20].

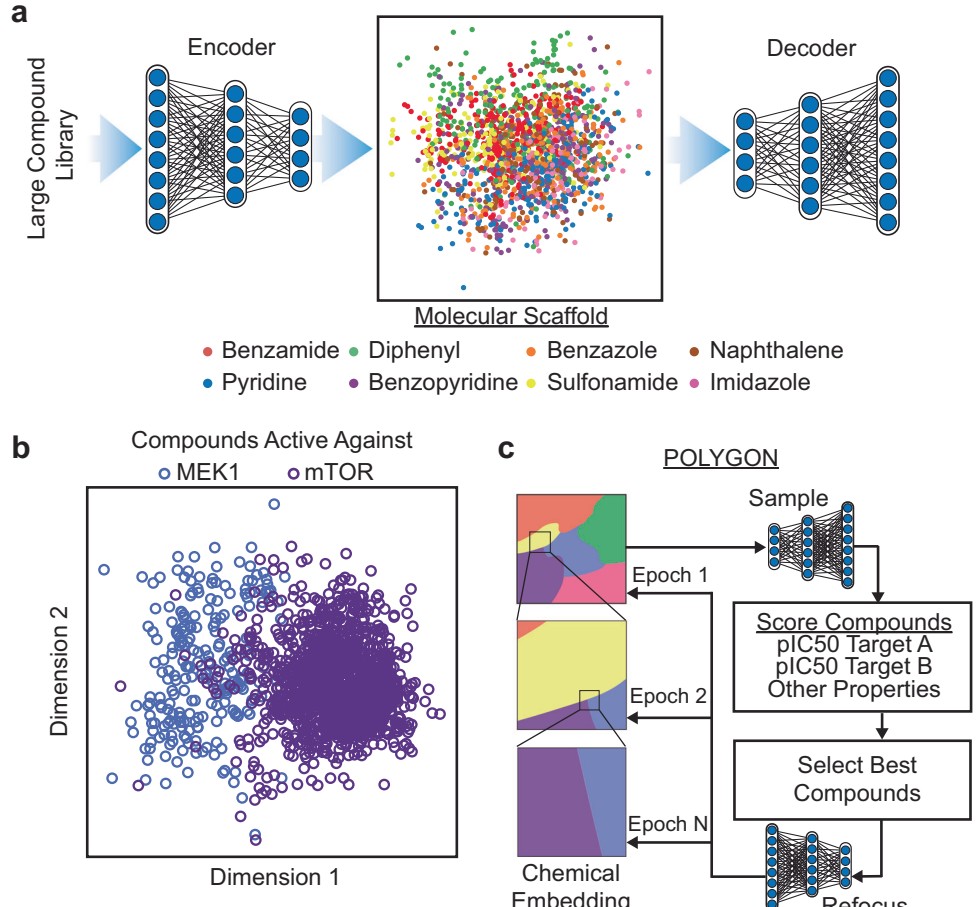

**Fig. 2 | Embedding chemical space for generative discovery of polypharmacology drugs. a** Use of a variational autoencoder (VAE) to create an embedded representation of chemical structure (middle), where input ChEMBL compounds (left) are encoded and decoded with separate deep neural networks. Here the embedding is approximated in two dimensions (Methods). Example embedded compounds are shown (points), along with their Murcko scaffold classification (colors). **b** MEK1 or mTOR-targeting compounds (blue or purple points) in the chemical embedding space, approximated in two dimensions as for (**a**).

**c** Reinforcement learning strategy for de novo generation of compounds recognizing two targets. Compounds are sampled from the chemical embedding (top) and scored by the predicted inhibitory concentration against each target (pIC50) alongside a panel of metrics to assess synthesizability and drug likeness (middle). Top-scoring compounds are used to refocus the chemical embedding for progressive epochs of compound sampling (descending arrows, bottom). Source data are provided as a Source Data file.

chi-squared test; Fig. 3b; performance at other IC50 thresholds shown in Fig. 3c, d).

### Generation of compounds targeting synthetic-lethal cancer proteins

We next asked POLYGON to perform de novo generation of polypharmacology compounds against ten pairs of protein targets that had been previously determined to be codependent (synthetic lethal) in human cancer cell lines (Fig. 4a)[31–33]. The targeted proteins comprised several protein classes, including serine/threonine kinases, tyrosine kinases, DNA binding factors, and histone modifiers. The top 100 highest scoring compounds for each target pair were selected. To model the interactions of these compounds against their corresponding protein targets, we performed molecular docking analysis using AutoDock Vina[34] and UCSF Chimera[35]. Across this set of compounds and targets, we found that the mean ΔG shift upon compound docking was favorable, at −1.09 kcal/mol, supporting the POLYGON predictions of binding (Fig. 4b $p = 9.25 \times 10^{-6}$; one-sided $t$ test = −4.285; DOF = 7146; 95% CI −1.21 to −0.98).

For example, to study POLYGON compounds targeting the synthetic-lethal combination of MEK1 (mitogen-activated protein

kinase kinase 1) and mTOR (mammalian target of rapamycin), we first queried the Protein Data Bank[36] to obtain the co-crystal structures of these proteins with their canonical single-target inhibitors. In particular, we obtained the structure of MEK1 with trametinib, a canonical MEK1 inhibitor, and of the mTOR-FRB/FKBP12 complex with rapamycin, a canonical mTOR inhibitor (PDB records 7M0Y and 3FAP). We verified that AutoDock Vina could correctly orient trametinib within MEK1 with a favorable ΔG of −9.2 kcal/mol, and that its best placement of trametinib within the second target, mTOR, was substantially less favorable at ΔG of −7.4 kcal/mol (Fig. 5a, b). In a complementary fashion, we confirmed that rapamycin was correctly placed in the mTOR complex with a favorable ΔG of −8.6 kcal/mol, and that its best placement within the first target, MEK1, was substantially less favorable with ΔG of −3.7 kcal/mol (Fig. 5c, d). We next investigated the docking positions of the top POLYGON compound (IDK12008), finding its best orientation in MEK1 to be similar to trametinib with a ΔG of −8.4 kcal/mol (Fig. 5e) and its best orientation in the mTOR complex to be similar to rapamycin with a binding energy of −9.3 kcal/mol (Fig. 5f). Qualitatively similar docking results were obtained for POLYGON-generated compounds against other synthetic-lethal protein pairs (Supplementary Fig. 4, Methods).

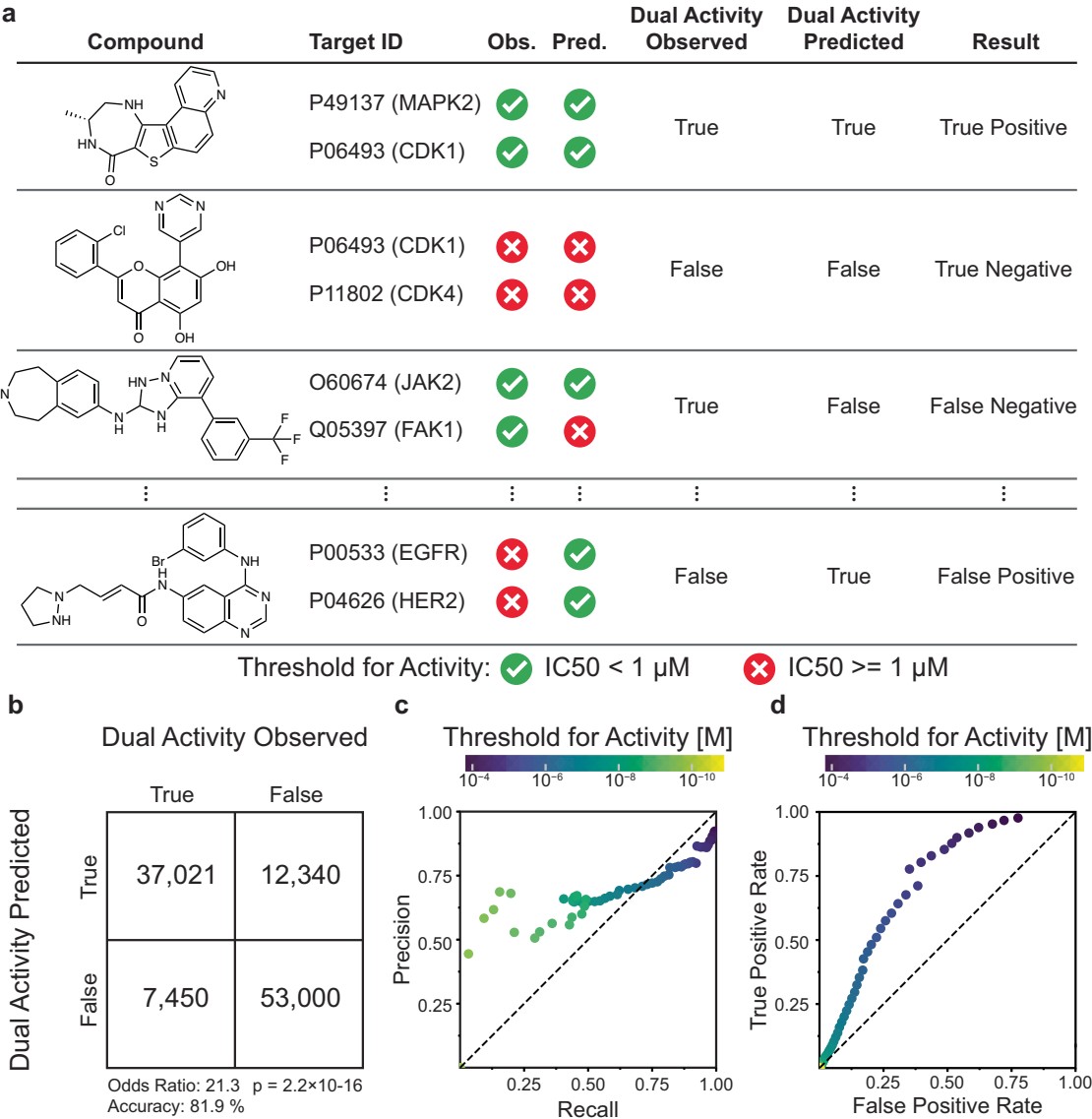

**Fig. 3 | Validation of specific compound-dual-target activities using POLYGON.**
**a** A library of 109,811 compounds was scored for activity against pairs of protein targets. (Compound, Target1, Target2) triplets for which low IC50 was observed against both targets (both activities <1 μM) were classified as "dual activity observed", otherwise as "dual activity not observed". Similarly, triplets for which low IC50 was predicted against both targets were classified as "dual activity predicted", otherwise as "dual activity not predicted". Observed and predicted dual activities were compared to tabulate true positive, false positive, false negative, and true negative cases. **b** Contingency table for classification of dual activity status of (Compound, Target1, Target2) triplets. Odds ratio computed by two-sided Fisher Exact Test and p-value computed by chi-squared test; exact *p*-values are provided in the Source Data file. **c** Precision-recall of POLYGON classification of dual active compounds at various IC50 activity thresholds (colored dots). **d** Receiver-operator characteristic of POLYGON classification of dual active compounds at various activity thresholds (colored dots). Source data are provided as a Source Data file.

## Synthesis and validation of dual MEK1/mTOR compounds

Given the current interest in MEK1 and mTOR kinases for combination therapy[37–40] (Supplementary Fig. 5a), we sought to validate POLYGON-generated compounds against these targets in the laboratory. First, we experimentally confirmed that current single-target inhibitors of MEK1 and mTOR can be combined to achieve a synergistic reduction of viability (Supplementary Fig. 5b–e); that this synergy extends to a wide variety of human cancer cells (Supplementary Fig. 5f); and that these effects are due to specific inhibition of each target (Supplementary Fig. 5g).

Turning to the top 100 de novo MEK1/mTOR candidate compounds, we synthesized 32 for validation, with the goal of minimizing reaction steps; as such, anilines were overrepresented due to shared synthetic routes (Supplementary Data 2, Supplementary Fig. 6). We first performed an activity screen of the synthesized compounds, including dose-response assays of overall cell growth inhibition (IC50, Fig. 6a) and direct measurements of phosphorylation activities of the target mTOR and MEK1 kinases (phospho-P70 and phospho-ERK, Fig. 6b, c). We found that the majority of compounds had IC50s in the 1–10 μM range (Fig. 6d) with a concomitant >50% reduction in both kinase activities (10 μM, Fig. 6e, f). Reductions in mTOR and MEK1 activity were significantly correlated with overall growth inhibition (Pearson's $\rho_{mTOR}$ = 0.47; DOF = 32; 95% CI 0.20 to 1.00; *p* = 0.0032; and $\rho_{MEK1}$ = 0.45; DOF = 32; 95% CI 0.17 to 1.00; *p* = 0.0049), suggesting that the molecular and cellular readouts were consistent. We further replicated the polyphamacologic capacity of six compounds with the largest reductions in mTOR and MEK1 activity in the primary screen, validating that four reduced phosphorylation activity of both targets by >50% at 1 μM (IDK12008, IDK12038, IDK12058, and IDK12065; 3 replicates; *p* < 0.05 by one-sided *t* test, Fig. 6g, Source Data). We also

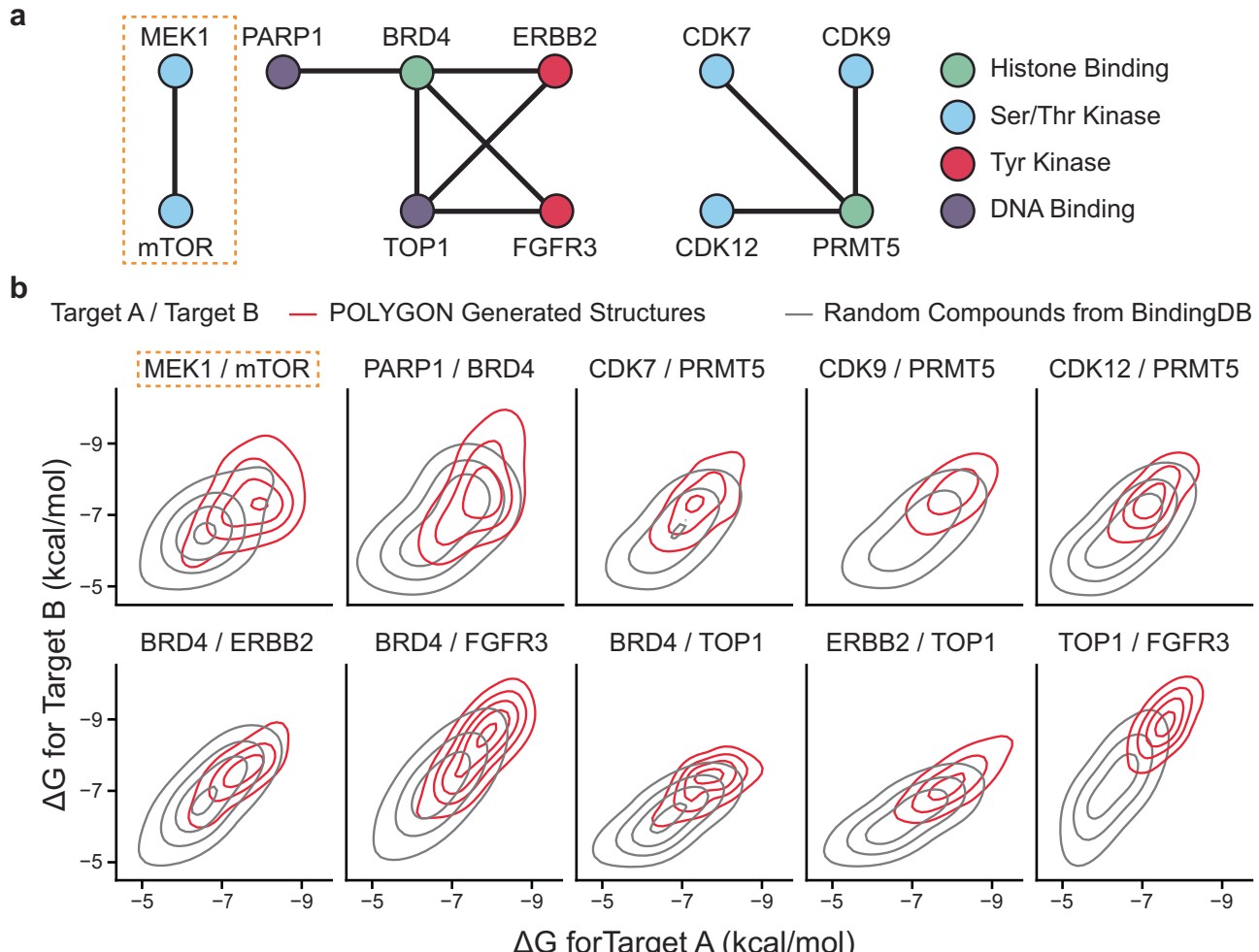

**Fig. 4 | Polypharmacology compounds targeting ten synthetic-lethal interactions. a** Synthetic-lethal interactions reported previously in human cancer cells (see text). Color corresponds to protein class: histone binding proteins are shown in green, serine/threonine kinases are blue, tyrosine kinases are red, and DNA binding proteins are purple. **b** Molecular docking analysis of POLYGON-generated compounds against each of the ten protein pairs from (**a**). Plots for each protein pair are arranged in a (5 × 2) grid. Each plot shows the distribution of binding activation energies for the docking configurations of 100 polypharmacology compounds generated by POLYGON (red) versus 100 randomly selected ligands from BindingDB (gray). For each target pair, the ΔG activation energies of POLYGON-generated compounds against each target were significantly more favorable (negative) than for compounds randomly drawn from BindingDB ($p < 1 \times 10^{-5}$ by one-sided $t$ test). Source data are provided as a Source Data file.

validated the inhibitory capacity of IDK12008 in a cell-free in vitro kinase assay, finding IC50 values for mTOR (Fig. 6h) and MEK1 (Fig. 6i) that were consistent with what was observed earlier in human cells (Fig. 6b, c, g).

Finally, we checked the specificity of the lead IDK compounds against mTOR/MEK1 by scoring the potential for off-target kinase inhibition. For this purpose, we used western blots to profile the phosphorylation activity of three representative unrelated kinases (PDK1, ATR, RAF) after exposure to each of the four doubly validated IDK compounds (Supplementary Fig. 7a–d). None of the representative kinases were reduced in activity by more than 20% by any IDK compound, with the exception of a 38% reduction in ATR activity after treatment with IDK12065. In addition to immunoblotting of these representative targets, we also performed a commercial screen of one of the lead IDK compounds, IDK12038, against a commercial panel of 371 human kinases (Methods). In these experiments, IDK12038 was shown to have activity against its target MEK1 (Supplementary Fig. 7e, Source Data); the other primary target, mTOR, was not included on the commercial panel. Otherwise, most kinases on the panel were minimally affected (330 out of 371 with >50% activity preserved post-treatment). This degree of promiscuity was similar to that of FDA-approved kinase inhibitors, which have been found to inhibit between 10 and 100 off-target kinases[41]. Regardless, one cannot rule out that, the effects of inhibiting some kinases on the kinome panel might lead to unintended cellular mechanisms that could be driving a portion of the effects seen in human cancer cell lines. This effect could explain the slight discrepancy in the potency between the biochemical and cell-based assays for mTOR (Fig. 6e, h).

In summary, our results demonstrate a pipeline by which candidate polypharmacology compounds are systematically generated, synthesized, and experimentally validated, resulting in a library of diverse molecular structures with activity against two targets. As presented, POLYGON addresses the initial phases of polypharmacology design, from which further optimization can proceed through classical techniques such as structure-activity relationships (SAR)[42]. It does not, in its current form, provide molecules optimized for absorption, distribution, metabolism, excretion, and toxicity (ADMET). An attractive avenue for further research would be to use SAR data collected from synthesized compounds for additional rounds of training to improve potency and selectivity against one or both targets. Such an iterative approach is akin to how medicinal chemists optimize compound structures following an initial molecular hit. Additionally, there are

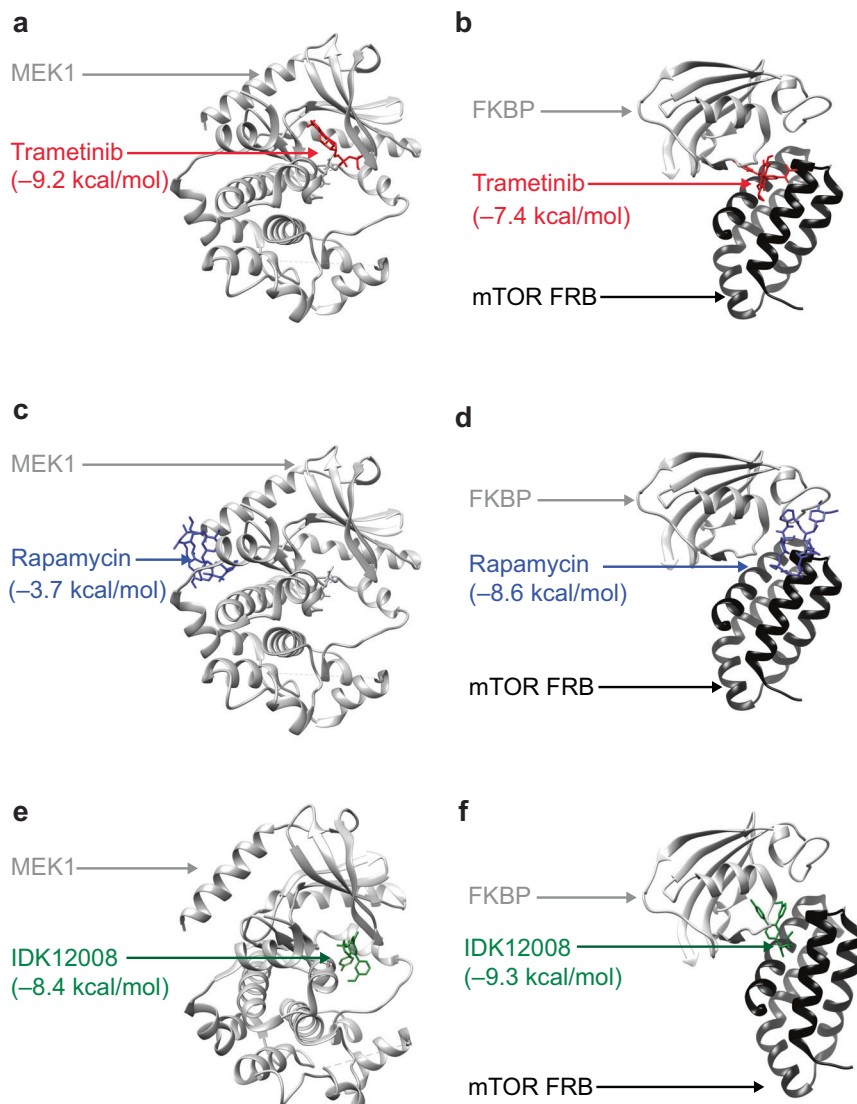

**Fig. 5 | Docking simulations of IDK compounds with mTOR and MEK1 structures. a** Lowest energy docking position of trametinib (red) in MEK1 crystal structure (gray). **b** Lowest energy docking position of trametinib (red) in the co-complex crystal structure of mTOR-FRB (black) and FKBP12 (gray). **c** Lowest energy docking position of rapamycin (blue) in MEK crystal structure (gray). **d** Lowest energy docking position of rapamycin (blue) in the co-complex crystal structure of mTOR-FRB (black) and FKBP12 (gray). **e** Lowest energy docking position of POLYGON generated compound IDK12008 (green) in MEK1 crystal structure. **f** Lowest energy docking position of POLYGON-generated compound IDK12008 (green) in co-complex crystal structure of mTOR-FRB (black) and FKBP12 (gray).

opportunities to improve the generative capacity of the POLYGON algorithm for dual protein inhibitors. Including IC50 predictions for a panel of off-target proteins (as a negative reward in the scoring module) could aid in minimizing unintended side effects. Likewise, including direct structural information of both the intended and unintended protein targets (again, in the scoring module) could allow for improvement in the potency of the generated small-molecule inhibitors. Given the proof-of-concept here, it will be exciting to explore polypharmacology compounds that exploit the growing number of genetic dependencies and synthetic-lethal combinations emerging from ongoing genomic and chemo-genomic screens, including the Cancer Dependency Map[43].

## Methods

### Variational autoencoder (VAE) architecture
Relevant to Fig. 2, Supplementary Fig. 1. POLYGON's VAE uses and extends code from the MOSES[17] and GuacaMol[15] pipelines for automated chemical design, which are based on deep learning modules in the PyTorch library (version 1.4.0). The VAE consists of two gated recurrent unit recurrent neural networks[17,44] (GRU-RNNs) implementing an encoder and decoder, respectively. The encoder unit $e(x) \rightarrow z$ converts any small molecule $x$ to a probability distribution $z$ of points in a chemical embedding (characterized by mean $\mu_z$ and standard deviation $\sigma_z$). The decoder unit $d(z) \rightarrow x'$ converts the embedding coordinates back into a valid small molecule, $x'$. During training, a VAE is optimized to minimize two different loss functions[18], one penalizing the reconstruction error, $x - d(e(x))$, and the other penalizing departures from normality (Kullback–Leibler divergence). To represent a small molecule $x$ numerically, the SMILES string of the molecule is padded so that strings for all molecules are of equal length=100, then converted to a floating point array of 128 dimensions using the PyTorch function torch.nn.Embedding. The encoder is then constructed as follows:

$$x \rightarrow F \rightarrow G \rightarrow \mu_z$$

$$\hookrightarrow G \rightarrow \sigma_z$$

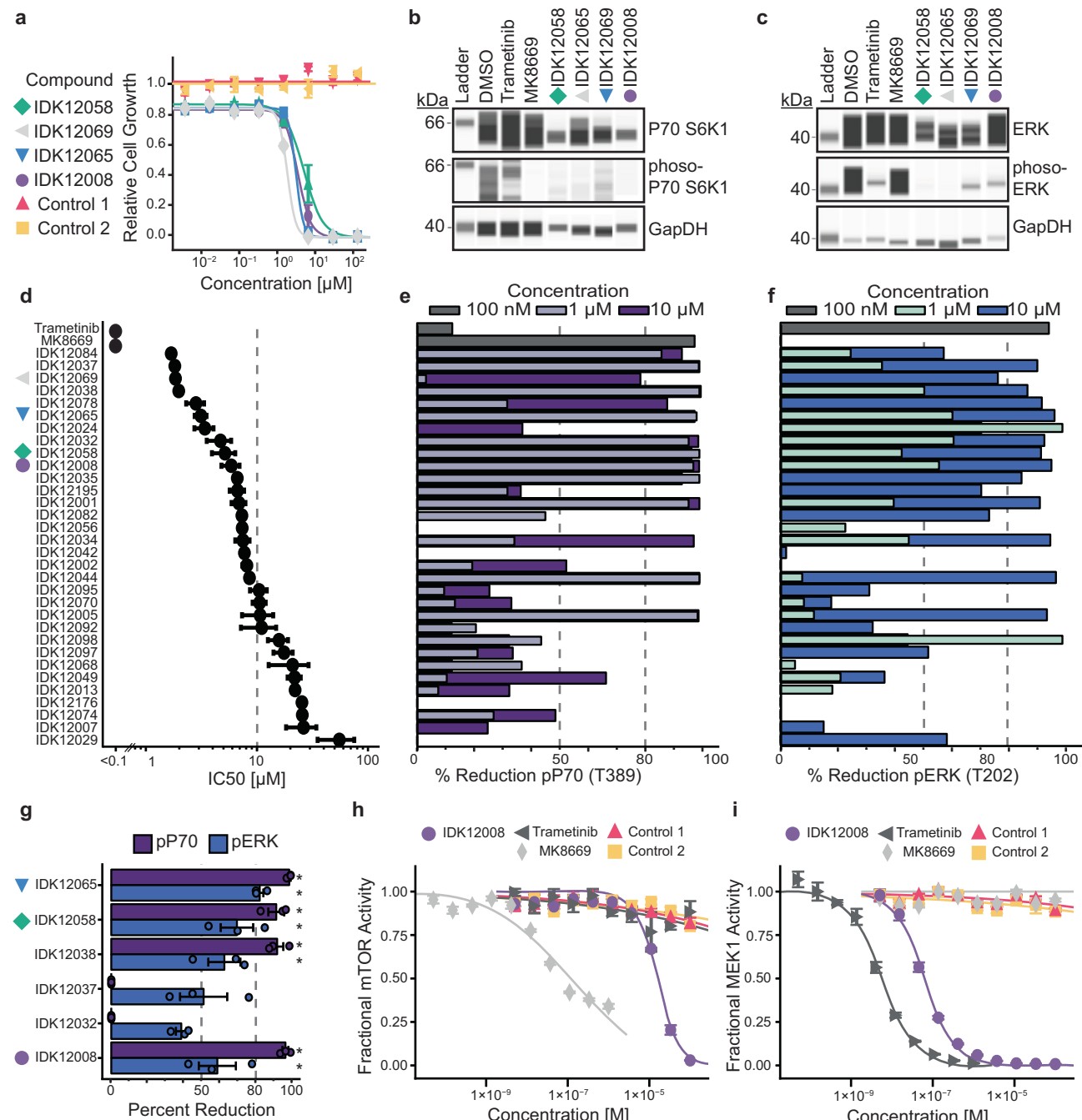

**Fig. 6 | Design and synthesis of de novo mTOR/MEK1 inhibitors. a** Growth of A549 cells (y-axis, relative to DMSO) treated with increasing concentrations of each of four compounds generated by POLYGON (IDK compounds) versus two negative controls (ChemBridge compounds 32574271 and 91530087, respectively). Error bars are the standard error of the mean cell growth across three independent replicates. **b** Capillary western blot of P70 S6K (top band), phospho-P70 S6K (middle band), and GAPDH (bottom band, loading control). Four synthesized IDK compounds are shown after treatment at 10 μM (vertical lanes) with MK-8669 treatment at 100 nM as the positive control and DMSO and trametinib treatment at 100 nM as the negative controls. Experiments were repeated three times with similar results. **c** Capillary western blot of ERK (top band), phospho-ERK (middle band), and GAPDH (bottom band, loading control). As for b, with trametinib as positive control and DMSO and MK-8669 as negative controls. Experiments were repeated three times with similar results. **d** IC50 from dose-response assays of IDK compounds in A549 cells. Error bars show the standard errors of the mean across at least three independent replicates. The first two rows (trametinib, MK-8669) are positive controls for MEK or mTOR inhibition, respectively. **e, f** mTOR (**e**) and MEK1 (**f**) activity measured by percent reduction of phospho-P70 S6K and phospho-ERK, respectively, upon three-hour treatment of IDK compounds at 1 μM or 10 μM. Controls were treated with MK8669 or trametinib for three hours at 100 nM (top dark gray bars). **g** Validation of MEK1 and mTOR activity upon three-hour treatment of IDK compounds at 1 μM, as measured by percent reduction of phospho-ERK and phospho-P70, respectively. Bars represent the mean across three independent treatments; error bars show the standard error of the mean; *$p < 0.05$ by one sample $t$ test. **h**, **i** Cell-free in-vitro mTOR/FRBP12 (**h**) and MEK1 (**i**) activity quantified with [33]P-labeled phosphate transfer upon three independent treatments of IDK compounds, select Chembridge compounds, trametinib, or MK-8669. Error bars are the standard error of the mean cell growth at a given dose across three independent replicates. Source data are provided as a Source Data file.

where $F()$ is torch.nn.GRU (1 layer, float output 256 dim, dropout rate of 0.2) and $G()$ is torch.nn.Linear (float output 128 dim). The decoder is constructed as $z = \mu_z + \varepsilon\ e^{(\sigma_z/2)}$, where $\varepsilon$ is drawn from a standard normal distribution ~ N(0,1).

$$z, x \;\downharpoonright$$

$$z \rightarrow H \rightarrow I \rightarrow J \rightarrow K \rightarrow x'$$

where $H()$ is torch.nn.Linear (float output 512); $I()$ is torch.nn.GRU with the output of H as the hidden state and the concatenation of $z$ and $x$ as an input (3 layers, float output 512 dim, dropout rate of 0.0); $J()$ is torch.nn.Linear (float output 55 dim); and $K()$ is the softmax function. Note that given the recurrent nature of the model, $x$ appears as an input to the decoder above. When sampling the chemical embedding without an input molecule, the input $x$ is replaced with a dummy array $w$. A SMILE sequence is then iteratively generated, character-by-character, updating $w$ after each character.

## Curation of small molecules in ChEMBL

Relevant to Fig. 2. The compound structures used for training and validation were obtained from a previously filtered set of molecules[15] from the ChEMBL 24 database[23]. Briefly, the ChEMBL database was processed by removing salts, charge neutralization, SMILES strings larger than 100 characters, and molecules containing atoms not in the set: {H, B, C, N, O, F, Si, P, S, Cl, Se, Br, I}. The resulting datasets were arbitrarily divided to form a total of 1,273,104 training molecules and 238,706 validation molecules.

## VAE training

Relevant to Fig. 2a, c, Supplementary Fig. 1. The VAE was trained using small molecules from the ChEMBL database as examples, as described in the main text. Training was performed across 200 epochs using the Adam optimizer[45] in PyTorch (learning rate of $3 \times 10^{-4}$, batch size of 1024 molecules, gradient clipping 50). The learned chemical embedding ($\mu_z$) was visualized by projecting into two dimensions with Principal Component Analysis (PCA) in scikit-learn v1.0.2. Original source code implementing the VAE is available on GitHub (https://github.com/bpmunson/polygon).

## Classifying compounds against protein kinase targets

Relevant to Fig. 2b, Supplementary Fig. 1d, 2. We queried the Pharos[27] GraphQL API and the BindingDB[25] for small molecule ligands against a list of 31 kinase proteins previously implicated in human cancer[24]. In concordance with the recommendations of the Pharos web interface, we selected ligands with an IC50 concentration of less than 1 μM against a given protein kinase target. We filtered the list of kinases to those with more than 300 ligands, resulting in the download of a total of 18,982 compounds each targeting one of 24 distinct protein kinases. SMILES strings for each compound were embedded using the VAE. The embedded values ($\mu_z$) of all compounds were projected into two dimensions with linear discriminant analysis, a supervised classifier, using the kinase targets as class labels (Python package scikit-learn, using the singular value decomposition solver SVD).

## Screening compounds

Two commercially available compounds were selected from the ChemBridge Core Library, IDs 32574271 and 91530087. The compounds were purchased from ChemBridge Corporation (www.chembridge.com, San Diego, CA).

## De novo molecular generation

Relevant to Figs. 2c, 4, 6; Supplementary Figs. 4, 6. To generate novel molecules with dual specificity to multiple targets (i.e. both mTOR and MEK1), the following reinforcement learning procedure was performed iteratively over 200 cycles: First, 8192 coordinates ($z$) from the chemical embedding were randomly sampled and decoded into molecular compounds (SMILES strings). Next, each of these compounds was scored against a set of six rewards $r_i$ as follows:

$r_1$, $r_2$: the predicted ligand efficiency[46] of the compound in binding each of the two protein targets. Further information about prediction of ligand efficiency is below in the next section.

$r_3$, $r_4$: the Euclidean distance of the compound's embedding ($\mu_z$) to the set of closest 20 known ligands for each of the two protein targets. From Pharos and BindingDB, see previous section.

$r_5$: the compound synthesizability (SA score)[47] computed using rdkit (version 2019.09.3)

$r_6$: the "drug likeness" of the compound (QED score)[30] also computed using rdkit.

The rewards $r$ were then each normalized ($r'$) to the range [0, 1], with 0 representing the worst performance and 1 the best. Normalization was achieved by half-Gaussian scaling, as per the GuacaMol[15] protocol:

$$r_i{'} = e^{-\frac{1}{2}((r_i - \mu_i)/\sigma_i)^2}$$

for $r_i < \mu_i$, otherwise 1 [for rewards $r_1$, $r_2$, $r_6$ which should be maximized]
for $r_i > \mu_i$, otherwise 1 [for rewards $r_3$, $r_4$, $r_5$ which should be minimized]

The threshold means ($\mu_i$) and standard deviations ($\sigma_i$) for normalization of each reward are provided in Supplementary Table 1. The normalized rewards were averaged to produce a single reward score $R$ for each molecule. Finally, the top-scoring 4096 molecules were used for additional training of the VAE (2 additional epochs, batch size 512 molecules, see above section on VAE training).

## Compound-target scoring module

Relevant to Figs. 2c, 3, Supplementary Fig. 3. Two random forest regression models (RFR, scikit-learn v1.0.2, 1000 trees) were constructed to predict ligand efficiency[46] of compounds generated to target MEK1 or mTOR, respectively. As training examples for each target, we collected ligand-target binding data from Pharos and BindingDB, resulting in 1146 or 5315 ligands with experimentally measured IC50 values against MEK1 or mTOR. These values were converted to ligand efficiency $y$:

$$y = 1.4 \left( \frac{-\log_{10} IC50}{N} \right)$$

where $N$ is the number of non-hydrogen atoms. To provide input features for the RFR model, each ligand was expressed as a 2048 bit Morgan fingerprint[48] (radius 2). Performance of the RFR models was measured with five-fold cross validation.

## Estimation of effect size

Relevant to Fig. 6d–f, Supplementary Fig. 3. Correlation coefficients were quantified with Pearson's correlation coefficient and Spearman's correlation coefficient in the python module 'scipy.stats' (version 1.11.3).

## Benchmarking against previous compound-target prediction methods

Relevant to Supplementary Fig. 3. While the focus of POLYGON is on the design of novel polypharmacology (dual target) compounds, many methods have been proposed to predict the affinity of existing

compounds against single kinase targets. In particular, the DREAM organization, in partnership with the Illuminating the Druggable Genome (IDG) program, recently held a drug-kinase binding prediction challenge which attracted a broad field of 268 submitted algorithms[12]. We thus reasoned that, while we were not aware of existing methods against which to benchmark the full POLYGON approach, its compound-target scoring module (Fig. 2c, above Methods) could be retroactively entered into the DREAM challenge and competed against the other algorithms. By the DREAM framework, predictions were scored by two complementary measures – Spearman rank correlation and root-mean-squared error – against the true binding affinity (Kd) values of 95 compounds measured for binding against 295 different kinase targets. Since the POLYGON compound-target scoring module was based on the predicted 50% inhibitory concentration (pIC50) rather than Kd (see above Methods section), we retrained its random forest regression (RFR) model to predict Kd, with all datasets and training procedures as described above for IC50. When used to predict the DREAM compound-kinase binding data, POLYGON predictions attained a Spearman rank correlation of 0.46 and 0.45 in Rounds 1 and 2 of the challenge, respectively (Supplementary Fig. 3a, b). This performance placed it in the top ~10% of all competing models (15 of 169 in Round 1 and 11 of 99 in Round 2), indicating the compound-target scoring component of POLYGON was competitive. The absolute top performing model in the challenge, developed by the team "AI Winter is Coming" (AIWIC), predicted binding inhibition from four different molecular fingerprints (radius os 5, 7, 9, 11) using 'xgboost'. We used this AIWIC model, available as a Docker container from Sage Bionetworks (syn15667962), to predict the relevant Kd values for all 100 IDK compounds which POLYGON had generated for dual mTOR and MEK1 binding. For the majority of these IDK compounds, AIWIC predicted dissociation constants of less than 1 μM for one of the targets and, for 20% of compounds, both targets (Supplementary Fig. 3c, d).

## Predicting polypharmacology of existing compounds

Relevant to Fig. 3. To quantify the ability of the compound-target scoring module in POLYGON, we tasked it with predicting whether a single compound was active against two specific protein targets. The BindingDB was filtered to compounds that were assayed against two and only two protein targets, resulting in 109,9811 compounds potentially polypharmacological compounds. We then filtered all of the candidate compounds from the BindingDB training dataset and used POLYGON to predict the individual compound-target IC50 values. Compound-target activities were categorized as active or non-active, where less than 1 μM IC50 value was defined to be active. Only compounds that had active predictions for both targets were classified as polypharmacological (Fig. 3a). POLYGON was able to predict the experimentally observed polypharmacology compounds with an Odds Ratio of 21.3 ($p$-value = $2.2 \times 10^{-16}$; 95% CI 20.7 to 22.0; chi-squared test; Fig. 3b). We also varied the threshold for active compound-target pairs, finding reasonable performance across a range of activity thresholds from $1 \times 10^{-5}$ M to $1 \times 10^{-10}$ M (Fig. 3c, d).

## Cell culture and reagents

Relevant to Fig. 6, Supplementary Fig. 5, 7. A549 cells were retrieved from the American Type Culture Collection (ATCC, CRM-CCL-185) and cultured in DMEM (Thermo Fischer Scientific, 11995065) + 10% FBS (Cell Culture Collective, Inc., FB-01). All cell lines tested negative for *Mycoplasma* contamination and were authenticated by short tandem repeat (STR) analysis. Trametinib (Selleckchem, S2673), MK-8669 (Selleckchem, S1022), molecules from ChemBridge, and de novo molecules from Bioblocks were dissolved in DMSO (10 mM, Sigma, D2650) and diluted in media for use. ChemBridge molecules were introduced to cells in the presence of 0.3 μL of lipofectamine (ThermoFisher, L3000150) to aid with cell permeability.

## Drug response and synergy determinations

Relevant to Fig. 6, Supplementary Fig. 5. Cell viability assays were conducted using the CellTiter-Glo Luminescent Cell Viability Kit (Promega, G7570) according to manufacturer specifications. Cells were seeded at 500 cells/well in a 384-well microtiter plate and grown for 24 h. At this time, compounds were added to the culture medium at the indicated concentrations (Fig. 6a, d, Supplementary Fig. 5b, d). Cells were then treated for 72 h before the addition of 25 μL CellTiter-Glo reagent, then analyzed on a Molecular Devices SpectraMax i3x. Single compound curves were analyzed using the neutcurve package in Python (version 0.5.7, https://jbloomlab.github.io/neutcurve/), after which drug combination effects were evaluated using the Loewe model of additivity[49] in the *synergyfinder* package[50] and plotted using *plotnine* (version 0.7.0).

## CRISPR-Cas9 gene knockouts

Relevant to Supplementary Fig. 5g. For gene knockout experiments, CRISPR-Cas9 nuclease was stably integrated in human A549 cells (ATCC, CRM-CCL-185) at the AAVS1 safe harbor locus. LentiCas9-Blast (Addgene plasmid # 52962; RRID:Addgene_52962) and lentiCRISPR v2 (Addgene plasmid # 52961; RRID:Addgene_52961) were gifts from Feng Zhang[51]. A549-Cas9 cells were tested for *Mycoplasma* contamination, expanded, then frozen in multiple aliquots so that experiments could be performed at low passage numbers. Cells were grown in DMEM (ThermoFischer, 11995065), 10% FBS (Cell Culture Collective, Inc., FB-01), and hygromycin (ThermoFischer, 10687010) to select for Cas9 expression, which was confirmed by capillary western (Wes, Protein Simple). Three unique 20-bp gRNAs were used for each target gene (Supplementary Table 2). The pooled library of double gRNA constructs (gene + gene or gene + non-targeting) was packaged into lentiviruses, and A549 cells were infected at an MOI of 0.3 to ensure each cell had zero or one double gRNA constructs. Puromycin selection (2.5 μg/mL, Sigma, P8833) was started two days after transduction, and the concentration was reduced by half upon each passaging, to a final concentration of 0.625 μg/mL, which was maintained for the remainder of the experiment. Following puromycin selection, cells were maintained in exponential growth by harvesting and removing a fraction of cells every two days. DNA was extracted from cells after 21 days of growth with a Blood and Cell Culture DNA Mini Kit (Qiagen, 13323) according to manufacturer protocols. To assess the frequency of gRNAs before and after selection, integrated DNA encoding the gRNA sequence was PCR amplified and prepared for HiSeq4000 sequencing (Illumina) according to manufacturer protocols. Standard Illumina primers were used for library preparation, and sequencing was conducted to generate 100-bp reads in a paired-end fashion. After sequencing, data quality was assessed with FastQC (0.11.9). The fitness effects of gene knockouts were determined as previously described[31] and normalized to the median fitness for non-targeting guides.

## Capillary immunoblotting

Relevant to Fig. 6, Supplementary Fig. 7. Lung cancer A549 cells (ATCC, CRM-CCL-185) were seeded into 6-well plates (400,000 cells/well) and treated 3 h at 1 μM or 10 μM for IDK compounds, 100 nM for MK-8669 (Selleckchem, S1022) and trametinib (Selleckchem, S2673), using DMEM (ThermoFischer, 11995065) as the diluent. At 3 h post-treatment, cells were collected by trypsinization (ThermoFisher, 25200114). Protein was isolated using M-PER Mammalian Protein Extraction Reagent (ThermoFisher, 78501) plus complete EDTA-free Protease Inhibitor Cocktail (Roche, 11873580001). Protein was quantified using 660 nM Protein Assay (Pierce, 1861426) with pre-diluted standard (Pierce, 23208) on a NanoDrop One spectrophotometer (ThermoFisher). Western blots were performed using capillary western (Wes, ProteinSimple). Protein was diluted to 3 mg/mL and run on a 12-230 kDa separation module (ProteinSimple, SM-W003) with an anti-

 

rabbit detection kit (ProteinSimple, DM-001). The expression of proteins of interest was measured using p44/42 MAPK (Erk1/2) (Cell Signaling Technology, 9102) (1:150), Phospho-p44/42 MAPK (Erk1/2) (Thr202/Tyr204) (Cell Signaling Technology, 9101) (1:500), p70 S6 Kinase (Cell Signaling Technology, 9202) (1:200), Phospho-p70 S6 Kinase (Thr389) (Cell Signaling Technology, 9205) (1:10), Phospho-AKT (Thr308) (Cell Signaling Technology, 9275) (1:100), Phospho-Chk1 (Ser345) (Cell Signaling Technology, 2348) (1:1000), Phospho-MEK1/2 (Ser217/221) (Cell Signaling Technology, 8154) (1:100) or GAPDH (14C10) Rabbit mAB (HRP conjugate) (Cell Signaling Technology, 3683) (1:4000). Band intensity was quantified as the area under the band peak with Compass for SW software (ProteinSimple, version 6.1.0) according to the manufacturer instructions. Uncropped images of the capillary immunoblots are provided in the Source Data.

### Compound synthesis
Relevant to Fig. 6, Supplementary Fig. 6. Synthetic planning and synthesis were carried out by Bioblocks Inc, San Diego CA. SMILES strings are provided for all molecules in Supplementary Data 1, synthesis pathways are provided for all molecules in Supplementary Data 2, and $^1$H NMR spectra are provided in Supplementary Data 3.

### Molecular docking simulations
Relevant to Figs. 4, 5, Supplementary Fig. 4. Simulations of the binding orientations of POLYGON-generated IDK compounds versus existing small molecule controls were performed using AutoDock Vina (version 1.1.2)[34] and UCSF CHIMERA (version 1.16)[35]. Receptor protein structures were extracted from X-ray diffraction structures in the Protein Data Bank (https://www.rcsb.org/). The structure of MEK1 and its small-molecule inhibitor trametinib was extracted from a larger structure which also contained the BRAF kinase in complex with AMP-PNP (PDB ID: 7M0Y). The FKBP12/FRB co-crystal structure was used to model the binding of rapamycin to mTOR within the MTORC1 complex (PDB ID: 3FAP). Likewise, we downloaded co-crystal structures for the receptor-ligand pairs of PARP1 in complex with olaparib (PDB ID: 7KK4) and BRD4 in complex with JQ1 (PDB ID: 3MXF). For the other protein targets we extracted the relevant chain from the following PCB IDs: CDK7 from 6XD3, CDK9 from 6Z45, CDK12 from 7NXK, PRMT5 from 6RLQ, ERBB2 from 7PCD, FGFR3 from 6LVM, and TOP1 from 1TL8. Docking positions were computed with default AutoDoc settings, with the exception of an exhaustive search setting of 8 and a maximum energy difference of 8 kcal/mol. Search volumes were set to encompass the entire crystal structure instead of specific a priori subdomains.

### In itro kinase binding
Relevant to Supplementary Figs. 7e. In-vitro whole kinome screening was performed with IDK12038 at 10 µM treatment and 1 µM ATP concentration against a panel of 371 wild-type human kinases using the HotSpot Assay (Reaction Biology Wild Type Panel) (data provided in Source Data).

### Cell-free kinase binding
Relevant to Fig. 6. In-vitro inhibition of mTOR (Fig. 6h) and MEK1 (Fig. 6i) activity by IDK12008, Trametinib, MK8669, and two negative control compounds (ChemBridge compounds 32574271 and 91530087) was measured by HotSpot Assay (Reaction Biology) across a three-fold serial dilution with three independent replicates. The mTOR inhibition assay included FKBP12 as a co-factor.

### Statistics and reproducibility
Relevant to Figs. 2, 3, 6 and Supplementary Fig. 5. We chose eight molecular scaffolds to highlight in the embedding of the chemical space (Fig. 2a) as a balance between highlighting chemical diversity and interpretability. To validate the specific compound-dual-target activities predicted by POLYGON we filtered the BindingDB to

compounds that had activity profiles measured against two and only two protein structures, this resulted in 109,811 unique compounds. We chose ten pairs of protein targets for compound generation to highlight the generalizability of POLYGON across different proteins and different protein classes. The number of compounds synthesized in this study, 32, was a result of minimizing the number of reaction steps used across all molecules.

In general, we chose conventional statistical analysis, such as Pearson's correlation coefficient (Fig. 6d–f), one-sided $t$-tests (Figs. 4b, 6g), and chi-squared tests (Fig. 3b). For statistical analysis with many groups (Fig. 6g), exact $p$-values are provided in the Source Data.

### Reporting summary
Further information on research design is available in the Nature Portfolio Reporting Summary linked to this article.

## Data availability
All datasets and materials generated in this study are provided in the Supplementary Information/Source Data or from the corresponding author on request. A key resource to the POLYGON framework is experimental binding data of small molecule ligands. We use the BindingDB (https://www.bindingdb.org/), ChEMBL 24 database (https://doi.org/10.6019/CHEMBL.database.24.1), and the Pharos (https://pharos.nih.gov/) as a source for this information. Molecule training datasets are available from the GuacaMol package: https://github.com/BenevolentAI/guacamol (https://doi.org/10.1021/acs.jcim.8b00839). Validation data of MEK1 and mTOR synergy under combination therapy across a panel of cancer cell lines was sourced from O'Neil et al.[52]. (https://doi.org/10.1158/1535-7163.mct-15-0843). For molecular docking simulations, we sourced the receptor protein structures from the Protein Data Bank (https://www.rcsb.org/). The following accession codes were used in this study: 7M0Y, 3FAP, 7KK4, 3MXF, 6XD3, 6Z45, 7NXK, 6RLQ, 7PCD, 6LVM, and 1TL8. Source data are provided with this paper.

## Code availability
The POLYGON source code is publicly available at https://github.com/bpmunson/polygon and https://doi.org/10.5281/zenodo.10712325[53].

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

## Acknowledgements

The authors gratefully acknowledge the support for this work provided by the NCI Cancer Cell Map Initiative (CA274502, OD032742) and other grants from the National Institutes of Health to T.I. (GM103504, ES014811), J.F.K. (CA243885) and B.M.K. (CA212456). We wish to thank Dr. Anthony Sun for his many valuable suggestions during the course of this study. We thank Dr. Peter Pallai and Valerie Laufer for their tremendous help and guidance in compound synthesis.

## Author contributions

B.P.M., B.M.K, J.K. and T.I. designed the study and developed conceptual ideas. B.P.M. and B.M.K. collected all the input sources and additional data. B.P.M. and B.M.K. implemented the main algorithm and all other computational methods and analyses. B.P.M., B.M.K., M.C., A.B. and K.L. performed all experiments and analyzed the results. B.M.K., B.P.M., R.A. and T.I. wrote the manuscript with suggestions from other authors.

## Competing interests

T.I. is a co-founder of Data4Cure, Inc. and has an equity interest. T.I. has an equity interest in Ideaya BioSciences, Inc. and is on the Scientific Advisory Board. T.I. is co-founder of Serinus Biosciences and has an equity interest. R.A. is a co-founder and a member of Molsoft LLC. The terms of this arrangement for T.I. and R.A. have been reviewed and approved by the University of California, San Diego, in accordance with its conflict of interest policies. The remaining authors declare no competing interests.
