## [Peer Review File · Nature Communications]

De novo generation of multi-target compounds using deep generative chemistryEditorial Note: This manuscript has been previously reviewed at another journal that is not operating a transparent peer review scheme. This document only contains reviewer comments and rebuttal letters for versions considered at *Nature Communications*.

Reviewer #1 (Remarks to the Author):

I thank the authors for a careful and comprehensive response to my previous comments. I think this is a strong manuscript and I am very satisfied with it as a nice contribution, filled with interesting innovations.

Reviewer #2 (Remarks to the Author):

The authors Munson, B. P. et al. present discussion of a computational tool, CHEMIST, a deep machine learning model they have developed to predict structures of polypharmacological compounds. In this context, "polypharmacological" refers to these compounds' ability to bind and elicit changes in activity of more than one target. The authors present a number of reasons for the utility of this type of compound in various disease states and proceed with using CHEMIST to generate compounds targeting both mTOR and MEK1, two Ser/Thr Kinases with cited effort to drug simultaneously. They then conduct molecular docking of CHEMIST-generated compounds against mTOR and MEK1 and select 32 for synthesis and activity assays to verify their [poly]pharmacological potential. The authors conclude by identifying 4 inhibitors ($IC_{50} < 1\mu M$) and profiling them for off-target effects on unrelated kinases. The work presented in this manuscript has potential to provide interesting insight on an approach to a generally unexplored area of drug development; however, it is seriously lacking analyses and crucial experimental data to warrant publication in Nat. Commun. Below are my concerns:

1. Generally, the authors have been receptive of the reviewers' comments and have made a number of modifications and additions to the original manuscript in an effort to address them. However, some of their responses do not meet my expectations. For example, the authors have copied and pasted a few of portions of their responses to some of the previous reviewers' comments, which seems to be a low-effort way of addressing common concerns among them. Also, the authors make a point to highlight the two years and \$200,000 of resources invested in this project in responses to reviewer comments – this is irrelevant and does not excuse the absence of important experimental data (KD profiling for compounds against the kinome is relatively inexpensive). They further cite a few other Nat. Commun. publications with much less or no experimental data; however, the scope of this project necessitates inclusion of the proper experimental validation of in silico predictions to fully support its conclusions.
2. The opening sentence of the authors' concluding paragraph reads "...candidate polypharmacology compounds are systematically designed..." I disagree that any of the compounds tested in any form were "designed," systematically or otherwise. More accurately, they are "generated," which is the language used in the Methods section, "De novo molecular generation." The authors should revise their use of the word "design" when used to refer to the function of their tool, especially in their manuscript title. This, unfortunately, diminishes the impact of this work, as the authors have not developed a tool for true de novo design of molecules, as they have claimed.
3. Previous reviewers made comments concerning the choice of two kinases to validate their approach to generating molecules which will target mTOR and MEK1, as both are kinases and are susceptible to inhibition by promiscuous agents. I appreciate the fact that the authors examined 9 other pairs of targets and used CHEMIST to generate new agents targeting their overlap, but this is largely meaningless without the same experimental validation of selected compounds done for mTOR and MEK1. I recognize the authors have spent a great deal of time and funding on this work already, but this feels like a crucial, missing piece to the study. The novelty of this study suffers greatly without more than one example of its functionality.
4. The authors discuss chemical embedding of compounds targeting various proteins, displayed in KDE plots. An overlap of two of these plots is used to show that polypharmacological drugs should exist in the chemical space between mTOR and MEK1. This is a reasonable conclusion.
 - a. However, the authors do not show these overlapping KDE plots for the rest of the 9 pairs of targets to validate approaching them for polypharmacological drug development or any further consideration.
 - b. Also, a good negative control is missing here. I would suggest demonstrating that the encoder distinguishes between distributions of compounds that do not bind to two unique targets. Show that unrelated targets bind different series of compounds in the same encoded space. This would be especially helpful for any of the targets paired with BRD4 in this analysis; it would support the

notion that the encoder distinguishes between the chemical space of compounds that bind those targets.

5. Figures are lacking counts for measured quantities (i.e. numbers of compounds analyzed $n = \#$).

6. In all of their experimental validation for compounds predicted to bind mTOR and MEK1, there is no presence of affinity data to demonstrate that the predicted compounds are actually binding their intended targets. The collected IC50 data could be indicative of inhibiting completely unrelated or unpredicted targets. A study within this scope should confirm that compounds intended for their targets do, indeed, bind their targets. Molecular docking and IC50 data do not suffice. Additionally, compounds with IC50s only slightly below μM are not considered 'potent' kinase inhibitors; at this concentration, a compound is likely to bind many other kinases. The authors do later profile one of their lead compounds IDK12038 against 371 kinases at 10 μM in Ext. Data Fig. 8f, but this is much too high of a concentration to demonstrate genuine potency or significant selectivity.

Unfortunately, the lack of the affinity data to confirm any of these predicted molecules as true binders represents perhaps the most serious flaw in this study.

7. In profiling their hit IDK12038 against 371 kinases, only MEK1 is indicated on Ext. Data Fig. 8f. I recognize that mTOR was missing from this panel. However, related to my previous point, it would be useful to see where targets not predicted to bind this compound fall on this curve (i.e. the other analyzed Ser/Thr kinases). If their encoder does not distinguish between targets their generated compounds bind or do not bind, this cuts into the novelty of this study.

Reviewer #3 (Remarks to the Author):

Overall, the authors have made several adjustments and improvements to the manuscript, including biochemical assays, broader kinase profiling and *in silico* prediction of new compounds with other dual targeting. However, the end result is that the novel compounds that have been synthesized are still largely mystery molecules in terms of their biological activities.

The kinome profiling experiment does not argue that MEK1 is one of the top targeted kinases. It appears to have about 40% residual activity using 10 μM concentration. Several other targets are much better inhibited. These kinome profiling results need to be provided (in the form of a table) so that the reader can assess the other kinases targeted by the profiled compound and understand what other targets the inhibitor(s). It is also highly unfortunate that the authors picked a kinome profiling product that does not contain mTOR. It would have been a lot more informative to have had data from a kinome profiling panel that includes mTOR. Now we are left to guess about the mTOR targeting and how it relates other kinase targets. Since the MEK1 is not among the very top targets of the tested broader panel, but the detailed data is not included, we also don't know what the top targets of this inhibitor are. It does also raise concerns that the effects seen in the cell-based assays (figure 6) are caused by inhibition of other kinases, especially since it appears that the effects in cells may occur at lower concentrations than the biochemical assay kinase inhibition occurs. (Unless all cell-based effects are due to mTOR inhibition, which biochemically remains an unknown).

There is no doubt that the approach and methodology is novel and exciting, and that the authors have put a significant amount of effort into the work. However the validation results do not convince this reviewer that compounds that are acting through dual MEK1/mTOR inhibition have been generated.

Manuscript NCOMMS-23-26224-T by Munson et al. “*De novo* generation of multi-target compounds using deep generative chemistry.”

Referee comments in black, our responses in blue. Key changes made to the manuscript are denoted by underlined blue text.

Reviewer #1 (Remarks to the Author):

I thank the authors for a careful and comprehensive response to my previous comments. I think this is a strong manuscript and I am very satisfied with it as a nice contribution, filled with interesting innovations.

We thank the reviewer for their positive assessment of our revisions and manuscript.

Reviewer #2 (Remarks to the Author):

The authors Munson, B. P. et al. present discussion of a computational tool, CHEMIST, a deep machine learning model they have developed to predict structures of polypharmacological compounds. In this context, “polypharmacological” refers to these compounds’ ability to bind and elicit changes in activity of more than one target. The authors present a number of reasons for the utility of this type of compound in various disease states and proceed with using CHEMIST to generate compounds targeting both mTOR and MEK1, two Ser/Thr Kinases with cited effort to drug simultaneously. They then conduct molecular docking of CHEMIST-generated compounds against mTOR and MEK1 and select 32 for synthesis and activity assays to verify their [poly]pharmacological potential. The authors conclude by identifying 4 inhibitors ($IC_{50} < 1\mu M$) and profiling them for off-target effects on unrelated kinases. The work presented in this manuscript has potential to provide interesting insight on an approach to a generally unexplored area of drug development; however, it is seriously lacking analyses and crucial experimental data to warrant publication in Nat. Commun. Below are my concerns:

1. Generally, the authors have been receptive of the reviewers’ comments and have made a number of modifications and additions to the original manuscript in an effort to address them. However, some of their responses do not meet my expectations. For example, the authors have copied and pasted a few of portions of their responses to some of the previous reviewers’ comments, which seems to be a low-effort way of addressing common concerns among them. Also, the authors make a point to highlight the two years and \$200,000 of resources invested in this project in responses to reviewer comments – this is irrelevant and does not excuse the absence of important experimental data (KD profiling for compounds against the kinome is relatively inexpensive). They further cite a few other Nat. Commun. publications with much less or no experimental data; however, the scope of this project necessitates inclusion of the proper experimental validation of in silico predictions to fully support its conclusions.

We appreciate the comment that time and monetary costs should not be driving factors in manuscript review, and we regret bringing these points up earlier as they appear to have

offended this reviewer. However, the fact remains that the experimental validation we have provided is substantially in excess of what has been provided in previous generative chemistry manuscripts. The most directly comparable manuscript to ours, focused on *in silico* drug generation, is Moret, M. *et al.* Leveraging molecular structure and bioactivity with chemical language models for *de novo* drug design. *Nat. Commun.* **14**, 1–12, 2023. In this study, the authors synthesize and validate two compounds against a single target and validate the structures with single-dose western blot analysis. Here, we provide 31 compounds experimentally assayed and provide multiple doses for the western blot analysis. Furthermore, specific to this revision, we have added several significant new datasets and analyses to support the approach, as we describe in response to the specific comments below.

2. The opening sentence of the authors' concluding paragraph reads "...candidate polypharmacology compounds are systematically designed..." I disagree that any of the compounds tested in any form were "designed," systematically or otherwise. More accurately, they are "generated," which is the language used in the Methods section, "De novo molecular generation." The authors should revise their use of the word "design" when used to refer to the function of their tool, especially in their manuscript title. This, unfortunately, diminishes the impact of this work, as the authors have not developed a tool for true *de novo* design of molecules, as they have claimed.

We appreciate that this reviewer feels AI/ML systems should not be said to "design" molecules in the same way people do. Given this feedback, we have changed the word "designed" to either "discovery" or "generated" throughout the manuscript. In this vein the title is changed to: *De novo generation [rather than design] of multi-target compounds using deep generative chemistry.* Furthermore, we have modified the name we gave our algorithm, "CHEMIST", as this name also suggests an AI/ML system which is acting to replicate human function and thus may offend the sensibilities of some readers. Instead, we now term the algorithm POLYGON, which stands for POLYpharmacology Generative Optimization Network. We believe this new name better captures the mechanism used to propose new compound structures.

3. Previous reviewers made comments concerning the choice of two kinases to validate their approach to generating molecules which will target mTOR and MEK1, as both are kinases and are susceptible to inhibition by promiscuous agents. I appreciate the fact that the authors examined 9 other pairs of targets and used CHEMIST to generate new agents targeting their overlap, but this is largely meaningless without the same experimental validation of selected compounds done for mTOR and MEK1. I recognize the authors have spent a great deal of time and funding on this work already, but this feels like a crucial, missing piece to the study. The novelty of this study suffers greatly without more than one example of its functionality.

In response to the earlier round of review, we had expanded our molecule generation work to cover nine additional pairs of proteins which recent studies have nominated as promising synergistic drug target combinations. Of these new pairs, PARP1-BRD4 and TOP1-BRD4 do not involve kinases, and for other pairs such as CDK7-PRMT5, one of the targets is a kinase and one is not. In all cases, molecular docking analysis indicates that the POLYGON system is able to generate compounds with energetically favorable recognition of both targets in the expected binding pockets.

To add further support for the POLYGON approach beyond MEK1/mTOR, we have extensively validated its ability to identify the multiple proteins targeted by each of more than 100,000 previously synthesized compounds with measured polypharmacology (new **Fig. 3**, reproduced below). This analysis shows that POLYGON correctly predicts the drug-target interactions of

polypharmacology compounds with ~82% accuracy, covering targets spanning a wide variety of protein classes not limited to kinases. Previously, some of this work had been mentioned but only very briefly in a supplemental figure panel; thus, we suspect it was either missed or unclear to reviewers. We clarified this analysis further by including additional relevant text and figures, forming the basis for the new main Figure 3.

Figure 3. Validation of specific compound-dual-target activities using POLYGON. **a**, A library of 109,811 compounds was scored for activity against pairs of protein targets. (Compound, Target1, Target2) triplets for which low IC₅₀ was observed against both targets (both activities < 1 μM) were classified as “dual activity observed”, otherwise as “dual activity not observed”. Similarly, triplets for which low IC₅₀ was predicted against both targets

were classified as “dual activity predicted”, otherwise as “dual activity not predicted”. Observed and predicted dual activities were compared to tabulate true positive, false positive, false negative, and true negative cases. **b**, Contingency table for classification of dual activity status of (Compound, Target1, Target2) triplets. Odds ratio and p-value computed by Fisher Exact Test. **c**, Precision-recall of POLYGON classification of dual active compounds at various IC50 activity thresholds (colored dots). **d**, Receiver-operator characteristic of POLYGON classification of dual active compounds at various activity thresholds (colored dots).

4. The authors discuss chemical embedding of compounds targeting various proteins, displayed in KDE plots. An overlap of two of these plots is used to show that polypharmacological drugs should exist in the chemical space between mTOR and MEK1. This is a reasonable conclusion. **a**. However, the authors do not show these overlapping KDE plots for the rest of the 9 pairs of targets to validate approaching them for polypharmacological drug development or any further consideration.

We now provide the requested KDE plots in an additional supplemental figure (**Extended Data Fig. 2**, reproduced below).

Extended Data Figure 2. Coembeddings in the chemical space for compounds targeting select protein kinases. Points represent compounds in BindingDB, colors label compounds determined to target a given protein target with $IC_{50} < 1 \mu M$. Dimensions 1 and 2 represent the two principal components of the common chemical embedding (Fig. 2a). Each panel superposes the compounds binding one target (row) with the compounds binding another (column). Compounds that are recognized by both targets are plotted twice.

b. Also, a good negative control is missing here. I would suggest demonstrating that the encoder distinguishes between distributions of compounds that do not bind to two unique targets. Show that unrelated targets bind different series of compounds in the same encoded space. This would be especially helpful for any of the targets paired with BRD4 in this analysis; it would support the notion that the encoder distinguishes between the chemical space of compounds that bind those targets.

The above KDE plots (new **Extended Data Fig. 2**) are meant to indicate that there might be accessible chemical space that can inhibit many different pairs of proteins. It does not make any attempt to imply that such a space does not exist for other pairs of protein targets.

Regardless, we think the newly presented analysis in **Figure 3** is relevant, as it characterizes the ability of POLYGON to predict the specific dual targets recognized by individual compounds. In this analysis across 109,811 compounds and 1,850 unique protein targets, we found POLYGON attained an accuracy of approximately 82% in classifying the specific targets for which a compound had polypharmacologic activity.

5. Figures are lacking counts for measured quantities (i.e. numbers of compounds analyzed $n = \#$).

As requested, we have added counts for all measured quantities in either the text or figures throughout the manuscript.

6. In all of their experimental validation for compounds predicted to bind mTOR and MEK1, there is no presence of affinity data to demonstrate that the predicted compounds are actually binding their intended targets. The collected IC₅₀ data could be indicative of inhibiting completely unrelated or unpredicted targets. A study within this scope should confirm that compounds intended for their targets do, indeed, bind their targets. Molecular docking and IC₅₀ data do not suffice. Additionally, compounds with IC₅₀s only slightly below μM are not considered 'potent' kinase inhibitors; at this concentration, a compound is likely to bind many other kinases. The authors do later profile one of their lead compounds IDK12038 against 371 kinases at 10 μM in Ext. Data Fig. 8f, but this is much too high of a concentration to demonstrate genuine potency or significant selectivity. Unfortunately, the lack of the affinity data to confirm any of these predicted molecules as true binders represents perhaps the most serious flaw in this study.

We have now incorporated the suggestion to provide *in-vitro* (cell-free) kinase activity profiling data for an AI-generated polypharmacology compound (IDK12008), alongside two positive controls (trametinib and MK8669) and two negative controls as suggested by the editors (ChemBridge compounds 32574271 and 91530087). Using this assay, we demonstrate the direct effects of IDK12008, a leading AI-generated polypharmacology compound, against MEK1 and mTOR. The new data are shown in **Fig. 6h-i** (reproduced below).

Figure 6. Design and synthesis of *de novo* mTOR/MEK1 inhibitors. h, Cell-free *in-vitro* mTOR/FRBP12 activity quantified with ^{33}P -labelled phosphate transfer upon treatment with IDK compounds, select Chembridge compounds, trametinib, and MK8669. i, Cell-free *in-vitro* MEK1 activity quantified with ^{33}P -labelled phosphate transfer upon treatment with IDK compounds, select Chembridge compounds, trametinib, and MK8669.

7. In profiling their hit IDK12038 against 371 kinases, only MEK1 is indicated on Ext. Data Fig. 8f. I recognize that mTOR was missing from this panel. However, related to my previous point, it would be useful to see where targets not predicted to bind this compound fall on this curve (i.e. the other analyzed Ser/Thr kinases). If their encoder does not distinguish between targets their generated compounds bind or do not bind, this cuts into the novelty of this study.

As discussed in the point above, we have now included *in-vitro* mTOR profiling of a candidate AI-generated compound (IDK12008) along with relevant control compounds. We agree with an earlier point made by this reviewer (comment #6 above) that a choice of 10 μ M concentration was much too high to demonstrate selectivity. Therefore, upon reflection we believe the new cell-free data, combined with the original off-target data we had generated for a few representative kinases (**Extended Data Fig. 7**, reproduced below) make for a more compelling argument.

Extended Data Figure 7. Assessment of IDK off-target kinase activity. a, PDK1 activity measured by AKT phosphorylation at Thr308 (pAKT, Cell Signaling #9275) upon treatment with 10 μ M IDK compounds or DMSO with GAPDH as control. b, ATR activity measured by CHK1 phosphorylation at Ser345 (Cell Signaling #2348) upon treatment with 10 μ M IDK compounds or DMSO with GAPDH as control. c, RAF1 activity measured by MEK1/2 phosphorylation at Ser217/221 (Cell Signaling #9154) upon treatment with 10 μ M IDK compounds or DMSO with GAPDH as control. d, Percent inhibition of kinases upon exposure to IDK compounds quantified from blots in panels a, b, and c. Activity quantified relative to GAPDH controls and DMSO-treated samples.

Reviewer #3 (Remarks to the Author):

Overall, the authors have made several adjustments and improvements to the manuscript, including biochemical assays, broader kinase profiling and *in-silico* prediction of new compounds with other dual targeting.

We thank the reviewer for their positive assessment of our revisions.

However, the end result is that the novel compounds that have been synthesized are still largely mystery molecules in terms of their biological activities. The kinome profiling experiment does not argue that MEK1 is one of the top targeted kinases. It appears to have about 40% residual activity using 10 μ M concentration. Several other targets are much better inhibited. These

kinome profiling results need to be provided (in the form of a table) so that the reader can assess the other kinases targeted by the profiled compound and understand what other targets the inhibitor(s).

A longstanding challenge in developing kinase inhibitors (not only ours) is their moderate promiscuity, and the results of our whole kinome screen are typical for FDA-approved kinase inhibitors (Hantschel, 2015). While the reviewer is correct that MEK1 was not the top kinase inhibited, it did come out in the top group of inhibited kinases. Thus we feel our kinome profiling experiment did indeed support the main conclusions of our study. Regardless, we agree that we had used a compound concentration that was quite high for this assay (10 μ M) and, as pointed out in the next reviewer comment below, the commercial kinome assays do not include the second drug target, mTOR. Therefore, upon reflection we believe the original off-target data we had generated for a few representative kinases (**Extended Data Fig. 7**, reproduced below) makes for a more compelling argument. We have thus now reverted to this figure.

Extended Data Figure 7. Assessment of IDK off-target kinase activity. a, PDK1 activity measured by AKT phosphorylation at Thr308 (pAKT, Cell Signaling #9275) upon treatment with 10 μ M IDK compounds or DMSO with GAPDH as control. b, ATR activity measured by CHK1 phosphorylation at Ser345 (Cell Signaling #2348) upon treatment with 10 μ M IDK compounds or DMSO with GAPDH as control. c, RAF1 activity measured by MEK1/2 phosphorylation at Ser217/221 (Cell Signaling #9154) upon treatment with 10 μ M IDK compounds or DMSO with GAPDH as control. d, Percent inhibition of kinases upon exposure to IDK compounds quantified from blots in panels a, b, and c. Activity quantified relative to GAPDH controls and DMSO-treated samples.

It is also highly unfortunate that the authors picked a kinome profiling product that does not contain mTOR. It would have been a lot more informative to have had data from a kinome profiling panel that includes mTOR. Now we are left to guess about the mTOR targeting and how it relates other kinase targets.

While the mTOR name implies ‘target of rapamycin’, its kinase activity is strongly dependent on physical interactions with several additional proteins including FKBP12. For this reason, mTOR is not included in the commercially available kinome panels. However, to address this comment we have now performed *in-vitro* (cell-free) kinase profiling against both drug targets (MEK1 and mTOR) for one of the polypharmacology compounds generated by POLYGON (IDE12008). These new results (**Figs. 6h,i**) indicate direct on-target effects and are presented alongside two positive controls (Trametinib and MK8669) and two negative controls, as suggested in conversation with the editors.

Figure 6. Design and synthesis of *de novo* mTOR/MEK1 inhibitors. **h**, Cell-free *in-vitro* mTOR/FRBP12 activity quantified with ^{33}P -labelled phosphate transfer upon treatment with IDK compounds, select Chembridge compounds, trametinib, and MK8669. **i**, Cell-free *in-vitro* MEK1 activity quantified with ^{33}P -labelled phosphate transfer upon treatment with IDK compounds, select Chembridge compounds, trametinib, and MK8669.

Since the MEK1 is not among the very top targets of the tested broader panel, but the detailed data is not included, we also don't know what the top targets of this inhibitor are. It does also raise concerns that the effects seen in the cell-based assays (figure 6) are caused by inhibition of other kinases, especially since it appears that the effects in cells may occur at lower concentrations than the biochemical assay kinase inhibition occurs. (Unless all cell-based effects are due to mTOR inhibition, which biochemically remains an unknown).

As discussed in the previous point, we now provide cell-free activity profiling of an AI-generated compound (IDE12008, Figs. 6h-i) to address the concern that the cell-based activities observed were a by-product of inhibition of other kinases.

There is no doubt that the approach and methodology is novel and exciting, and that the authors have put a significant amount of effort into the work. However the validation results do not convince this reviewer that compounds that are acting through dual MEK1/mTOR inhibition have been generated.

We again thank the reviewer for acknowledging the novelty and excitement of the approach.

Reviewer #3 (Remarks to the Author):

Ideker and colleagues have now made additional changes to address concerns raised by two of the reviewers.

This reviewer asked about providing better biochemical evidence for that the generated inhibitors (in particular the most tested inhibitor IDK12008) are indeed dual MEK/MTOR inhibitors and with what level of selectivity. This has partially been explored, but the overall level of information has been downgraded in this revised version. In the previous version of the manuscript, a kinome-wide profiling had been done, but the full data was not provided (the full results should of course have been provided). Surprisingly, this data has now been removed altogether, which clearly weakens the level of profiling provided and the understanding of the target spectrum of the compound. Yes, the 10 μ M concentration used in the kinome profiling was perhaps high, but it would have given an idea of the overall target spectrum and the polypharmacological nature of this compound. To instead show the effects on only three other cellular phosphorylation targets does not give a good idea of selectivity of the compound. Reducing the level information is not a way to achieve greater insight!

As a side note, MTOR was also not included in the previous kinome panel and the authors now claim that MTOR, because it needs accessory proteins for kinase activity is not part of commercial kinome panels. This is not true. Most (if not all) major kinome profiling service providers also have an MTOR assay (typically an mTORC1 assay) in their panels.

As the authors state in the rebuttal, many kinase inhibitors have several targets, but it is critical to understand what the targets are to be able to assess whether the intended primary target (or targets in this case) are in fact the main targets that cause cellular effects and has been a standard control procedure for kinase inhibitors for the last couple of decades.

The prototype MEK and MTOR (or more correctly mTORC1) inhibitors used in this study (trametinib, PD-325901 and MK8669/ridaforolimus) are exceptionally selective allosteric inhibitors, but the new dual inhibitors described here are presumably more likely ATP-competitive inhibitors (has this been tested?) and therefore likely more prone to be promiscuous. It would also mean that the new inhibitors are mTORC1 and mTORC2 inhibitors; very different than the indirect/allosteric mTORC1 inhibitor ridaforolimus that was used as a control. The partially described kinome profiling of IDK12008 in the previous version also suggested a relatively broad target spectrum. What that target spectrum is would be a critical part of the evaluation of these predicted dual inhibitors. It's good to know that it's not just an ultra-broad spectrum inhibitor that has been identified. The more selective inhibitors POLYGON can identify, the more useful it will be.

At a minimum. Including the previous kinome profiling (revealing the full data) is necessary, and if the target spectrum indeed looks broad, claims of the power of POLYGON identifying valuable inhibitors need to be done with caution, admitting that improvements in identifying more selective inhibitors can still be made. Even with these potential weaknesses, the POLYGON algorithm is interesting and publication-worthy. It's just important that it is fully tested, especially since the authors have gone through the efforts of synthesizing the predicted molecules and the biochemical validation is simple. Ideally, follow-up validation (dose-response testing) of the activities on "hits" from the kinome should be done and would strengthen the manuscript, but this is not absolutely essential.

There are some discrepancies between biochemical and cellular activities, especially for MTOR, where the potency in the cell-based testing appears significantly greater than in the biochemical testing. This is unusual, and could hint at that in the cells, an indirect mechanism occurs where one or more other kinases are inhibited at lower concentrations than MTOR, which in turn leads to inhibition of mTORC1 activity. This issue would be important to comment on, and including the kinome-wide profiling may also provide insight into what is happening.

In conclusion, the full kinome profiling dataset for IDK12008 that was referred to in the previous version needs to be provided and discussed. Depending on the results and possible level of further validation, conclusions and discussion needs to address to what extent the POLYGON algorithm can

generate reasonably selective inhibitors (and perhaps what could be improved in the future to sharpen identification of selective inhibitors).

Reviewer #3 (Remarks on code availability):

N/A

Manuscript NCOMMS-23-26224-T by Munson et al. “*De novo* generation of multi-target compounds using deep generative chemistry.”

Referee comments in black, our responses in blue. Key changes made to the manuscript are denoted by underlined blue text.

Reviewer #3 (Remarks to the Author):

Ideker and colleagues have now made additional changes to address concerns raised by two of the reviewers.

This reviewer asked about providing better biochemical evidence for that the generated inhibitors (in particular the most tested inhibitor IDK12008) are indeed dual MEK/MTOR inhibitors and with what level of selectivity. This has partially been explored, but the overall level of information has been downgraded in this revised version. In the previous version of the manuscript, a kinome-wide profiling had been done, but the full data was not provided (the full results should of course have been provided). Surprisingly, this data has now been removed altogether, which clearly weakens the level of profiling provided and the understanding of the target spectrum of the compound. Yes, the 10 μM concentration used in the kinome profiling was perhaps high, but it would have given an idea of the overall target spectrum and the polypharmacological nature of this compound. To instead show the effects on only three other cellular phosphorylation targets does not give a good idea of selectivity of the compound. Reducing the level information is not a way to achieve greater insight!

We appreciate the reviewer's request to provide the whole kinome panel that was included in the previous revision. As such, we have added back the relevant figure (new **Extended Data Fig. 7e**) as well as included the underlying raw data as a supplemental file (**Extended Data File 4**). Additionally, we have added relevant text to the Results and Discussion section, as follows:

Synthesis and validation of dual MEK1/mTOR compounds (Line number 175, page 8):

“In addition to immunoblotting of these representative targets, we also performed a commercial screen of one of the lead IDK compounds, IDK12038, against a commercial panel of 371 human kinases (Methods). In these experiments, IDK12038 was shown to have activity against its target MEK1 (**Extended Data Fig. 7e, Extended Data File 4**); the other primary target, mTOR, was not included on the commercial panel. Otherwise, most kinases on the panel were minimally affected (330 out of 371 with >50% activity preserved post-treatment). This degree of promiscuity was similar to that of FDA-approved kinase inhibitors, which have been found to inhibit between 10 and 100 off-target kinases⁴¹. “

As a side note, MTOR was also not included in the previous kinome panel and the authors now claim that MTOR, because it needs accessory proteins for kinase activity is not part of commercial kinome panels. This is not true. Most (if not all) major kinome profiling service providers also have an MTOR assay (typically an mTORC1 assay) in their panels.

Upon request the kinome profiling service included the necessary co-factor, FRBP12, for the mTORC1 complex in a single kinase assay. The results of this assay are provided in **Fig. 6h**.

As the authors state in the rebuttal, many kinase inhibitors have several targets, but it is critical to understand what the targets are to be able to assess whether the intended primary target (or targets in this case) are in fact the main targets that cause cellular effects and has been a standard control procedure for kinase inhibitors for the last couple of decades.

The prototype MEK and MTOR (or more correctly mTORC1) inhibitors used in this study (trametinib, PD-325901 and MK8669/ridaforolimus) are exceptionally selective allosteric inhibitors, but the new dual inhibitors described here are presumably more likely ATP-competitive inhibitors (has this been tested?) and therefore likely more prone to be promiscuous. It would also mean that the new inhibitors are mTORC1 and mTORC2 inhibitors; very different than the indirect/allosteric mTORC1 inhibitor ridaforolimus that was used as a control. The partially described kinome profiling of IDK12008 in the previous version also suggested a relatively broad target spectrum. What that target spectrum is would be a critical part of the evaluation of these predicted dual inhibitors. It's good to know that it's not just an ultra-broad spectrum inhibitor that has been identified. The more selective inhibitors POLYGON can identify, the more useful it will be.

We agree with the reviewer's suggestion to profile the effects on unintended targets and include the whole-kinome panel (**Extended Data Fig. 7e**) to address the issue.

At a minimum. Including the previous kinome profiling (revealing the full data) is necessary, and if the target spectrum indeed looks broad, claims of the power of POLYGON identifying valuable inhibitors need to be done with caution, admitting that improvements in identifying more selective inhibitors can still be made. Even with these potential weaknesses, the POLYGON algorithm is interesting and publication-worthy. It's just important that it is fully tested, especially since the authors have gone through the efforts of synthesizing the predicted molecules and the biochemical validation is simple. Ideally, follow-up validation (dose-response testing) of the activities on "hits" from the kinome should be done and would strengthen the manuscript, but this is not absolutely essential.

We do agree that improvement in identifying more selective inhibitors can and should still be made. As the reviewer states, we feel that POLYGON is a useful tool for generating *initial* chemistry targeting multiple protein targets. To reflect this point we have modified the Conclusion section, clarifying the opportunity for improvement:

Conclusion (Line number 187, page 9):

"Our results demonstrate a pipeline by which candidate polypharmacology compounds are systematically generated, synthesized and experimentally validated, resulting in a library of diverse molecular structures with activity against two targets. As presented, POLYGON addresses the initial phases of polypharmacology design, from which further optimization can proceed through classical techniques such as structure-activity relationships (SAR)⁴². It does not, in its current form, provide molecules optimized for absorption, distribution, metabolism, excretion, and toxicity (ADMET). An attractive avenue for further research would be to use SAR data collected from synthesized compounds for additional rounds of training to improve potency and selectivity against one or both targets. Such an iterative approach is akin to how medicinal chemists optimize compound structures following an initial molecular hit. Additionally, there are opportunities to improve the generative capacity of the POLYGON algorithm for dual protein inhibitors. Including IC₅₀ predictions for a panel of off-target proteins (as a negative reward in the scoring module) could aid in minimizing unintended side effects. Likewise, including direct structural information of both the intended and unintended protein targets (again, in the scoring module) could allow for improvement in potency of the generated small-molecule inhibitors. Given

the proof-of-concept here, it will be exciting to explore polypharmacology compounds that exploit the growing number of genetic dependencies and synthetic-lethal combinations emerging from ongoing genomic and chemo-genomic screens, including the Cancer Dependency Map⁴³.”

There are some discrepancies between biochemical and cellular activities, especially for MTOR, where the potency in the cell-based testing appears significantly greater than in the biochemical testing. This is unusual, and could hint at that in the cells, an indirect mechanism occurs where one or more other kinases are inhibited at lower concentrations than MTOR, which in turn leads to inhibition of mTORC1 activity. This issue would be important to comment on, and including the kinome-wide profiling may also provide insight into what is happening.

We have added two sentences to the Results and Discussion commenting on the differences between the biochemical and cellular activities:

Synthesis and validation of dual MEK1/mTOR compounds (Line number 182, page 9):

“However, the effects of inhibiting some kinases on the kinome panel might lead to unintended cellular mechanisms that could be driving a portion of the effects seen in human cancer cell lines. This effect could explain the slight discrepancy in the potency between the biochemical and cell-based assays for mTOR (**Fig. 6e,h**).”

In conclusion, the full kinome profiling dataset for IDK12008 that was referred to in the previous version needs to be provided and discussed. Depending on the results and possible level of further validation, conclusions and discussion needs to address to what extent the POLYGON algorithm can generate reasonably selective inhibitors (and perhaps what could be improved in the future to sharpen identification of selective inhibitors).

Reviewer #3 (Remarks to the Author):

The authors have now nicely addressed all the points raised by this reviewer and it now appears suitable for publication in Nature Communications.